# A Small Natural Molecule S3 Protects Retinal Ganglion Cells and Promotes Parkin-Mediated Mitophagy against Excitotoxicity

**DOI:** 10.3390/molecules27154957

**Published:** 2022-08-04

**Authors:** Dongli Zhuang, Rong Zhang, Haiyang Liu, Yi Dai

**Affiliations:** 1Department of Ophthalmology & Visual Science, Eye & ENT Hospital, Shanghai Medical College, Fudan University, Shanghai 200031, China; 2NHC Key Laboratory of Myopia (Fudan University), Key Laboratory of Myopia, Chinese Academy of Medical Sciences, and Shanghai Key Laboratory of Visual Impairment and Restoration (Fudan University), Shanghai 200031, China; 3The State Key Laboratory of Phytochemistry and Plant Resources in West China, Kunming Institute of Botany, Chinese Academy of Sciences, Kunming 650204, China

**Keywords:** excitotoxicity, retinal ganglion cells, 15-oxospiramilactone, mitophagy, parkin

## Abstract

Glutamate excitotoxicity may contribute to retinal ganglion cell (RGC) degeneration in glaucoma and other optic neuropathies, leading to irreversible blindness. Growing evidence has linked impaired mitochondrial quality control with RGCs degeneration, while parkin, an E3 ubiquitin ligase, has proved to be protective and promotes mitophagy in RGCs against excitotoxicity. The purpose of this study was to explore whether a small molecule S3 could modulate parkin-mediated mitophagy and has therapeutic potential for RGCs. The results showed that as an inhibitor of deubiquitinase USP30, S3 protected cultured RGCs and improved mitochondrial health against NMDA-induced excitotoxicity. Administration of S3 promoted the parkin expression and its downstream mitophagy-related proteins in RGCs. An upregulated ubiquitination level of Mfn2 and protein level of OPA1 were also observed in S3-treated RGCs, while parkin knockdown resulted in a major loss of the protective effect of S3 on RGCs under excitotoxicity. These findings demonstrated that S3 promoted RGC survival mainly through enhancing parkin-mediated mitophagy against excitotoxicity. The neuroprotective value of S3 in glaucoma and other optic neuropathies deserves further investigation.

## 1. Introduction

The progressive loss of retinal ganglion cells (RGCs) is a common feature of various optic neuropathies, while glutamate excitotoxicity has been considered to play an important role in the mechanism of RGC degeneration [1]. Accumulating evidence has linked impaired mitochondrial quality control in RGCs under stressful conditions [2,3]. Selective elimination of damaged mitochondria via a process known as mitophagy is essential for mitochondrial quality control. The E3 ubiquitin ligase parkin plays a vital role in mediating the canonical mitophagy pathway [4,5,6]. We previously found that genetically upregulating parkin protects RGCs against excitotoxicity and ameliorated the mitophagy defects [7]. It is thus of great interest to explore small molecules that could protect RGCs through the parkin-mediated mitophagy pathway.

Recent studies have shown that several deubiquitinases can counteract parkin-mediated mitochondrial ubiquitination and may serve as a key negative regulator of mitophagy. A broad spectrum deubiquitinase inhibitor, PR-619, has been reported to exert a neuroprotective effect and promote parkin-mediated mitophagy on RGCs against glutamate excitotoxicity [8]. Several lines of evidence have suggested that deubiquitinases USP30 could act as a potential target for enhancing mitophagy in both cellular and animal models of Parkinson’s diseases [4,6,9]. However, the effect of USP30 inhibition on RGCs under stressful environments remains elusive.

The identity of S3 is 15-oxospiramilactone, a small compound (molecular weight of 330 Da) derived from diterpenoids. Diterpenoids are atisine-type natural products obtained from the complex of Spiraea japonica, a Chinese medicine widespread in Yunnan Province of China [10]. It has been reported that S3 could act as a covalent inhibitor with a cysteine residue of the protein forming an adduct with the N-cyano group to block the deubiquitinases activity of USP30. Moreover, S3 may induce the non-degradative ubiquitination of mitofusin (Mfn), which promotes mitochondrial fusion in mouse embryonic fibroblasts [11]. However, little is known about the role of S3 in RGCs under stressful conditions. Thus, the purpose of the present study was to address whether the small natural molecule S3 could protect RGCs and its mechanism involved in regulating mitochondrial quality control under NMDA-induced excitotoxicity.

## 2. Results

### 2.1. S3 Protected RGCs from NMDA-Induced Excitotoxicity

To screen for optimal concentration of S3, we incubated cultured RGCs with different concentrations for 24 h, then assessed cell viability and mitochondrial membrane potential. According to the results of LDH and JC-1assays (*p* < 0.05; Figure 1A,B), the optimal concentration of S3 for RGCs was 2 μM, which was chosen for further experiments.

Compared with vehicle-treated blank RGCs, the percentage of apoptotic cell death was decreased (*p* < 0.01, Figure 1C) and the JC-1 ratio was increased (*p* < 0.01, Figure 1D) in S3-treated RGCs. LDH assay exhibited no significant differences (*p* > 0.05, Figure 1E). The results indicated that 2 μM S3 exhibited no detrimental effect on RGC viability.

In vehicle-treated RGCs under excitotoxicity as a control group, the percentage of apoptotic cell death was 29.39%, while the S3-treated RGCs showed significantly less apoptotic cell death (16.39%, *p* < 0.01; Figure 1C). Cytotoxicity and JC-1 ratio showed the same trend. S3 treatment increased JC-1 ratio by 6.79% and decreased the LDH activity of RGCs by 30.32% (*p* < 0.01; Figure 1D,E). These results suggested that S3 can protect RGCs under NMDA-induced excitotoxicity.

### 2.2. S3 Improved Mitochondrial Morphology and Mitophagy in RGCs under Excitotoxicity 

Double immunostaining with MitoTracker and LAMP1 or LC3 was performed on RGCs. In vehicle-treated blank RGCs, a tubular mitochondrial network was distributed around the cell body and axon. No significant change in mitochondria morphology, LAMP1 expression and their colocalization was observed in S3-treated blank RGCs (Figure 2A,B). Decreased immunoreactivity of LC3 was observed in S3-treated blank RGCs (Figure 2E,F). In NMDA-induced excitotoxic conditions, small dot-like mitochondria were dispersed in the soma and axon in vehicle-treated RGCs. Elongated mitochondria with increased amounts were present around the nucleus and in the axon, and a tubular network was partially restored in S3-treated RGCs (Figure 2C,D,G,H). These data indicated that S3 treatment reduced NMDA-induced mitochondrial fragmentation in RGCs. Meanwhile, increased immunoreactivity of LAMP1 and LC3, as well as their colocalizations with mitochondria in the axons, was observed in S3-treated RGCs under excitotoxicity (Figure 2I–L). 

RGCs were further co-stained with mitochondrial outer membrane protein TOMM20 and parkin. The immunoreactivity of parkin was upregulated, while TOMM20 remained the same under both S3-treated blank and excitotoxic conditions (Figure 3A–D,I,J). It is noteworthy that S3 treatment increased the colocalization of parkin with TOMM20 (Figure 3D,H,K), which indicated that S3 facilitated the parkin recruitment to mitochondria in RGCs under excitotoxicity.

Moreover, compared with the vehicle-treated controls, the level of isoforms L and S of OPA1 protein was upregulated (*p* < 0.01; Figure 4A,B), while no significant difference was observed for Mfn2 in S3-treated RGCs under both S3-treated blank and excitotoxic conditions (*p* > 0.05; Figure 4A,C). Then we performed immunoprecipitation to analyze the ubiquitination level of Mfn2. The results showed that compared with the control group, S3 treatment significantly increased the ubiquitination level of Mfn2 in RGCs under excitotoxicity (Figure 4D).

### 2.3. Parkin Knockdown Reduced the Effect of S3 on RGCs under Excitotoxicity 

To investigate the mechanism associated with the S3 action on mitophagy, RGCs were transfected with lipidosome-carrying parkin siRNA to knockdown its expression. The control group was transfected with lipidosome carrying si-Null. Both groups received S3 treatment under NMDA-induced excitotoxicity.

Western blot analysis confirmed that the expression of parkin protein was decreased by approximately 50% in RGCs in the si-Parkin group (*p* < 0.01; Figure 5A,E). Compared with the control group, the cytotoxicity as reflected by LDH release was significantly increased by 201.1% in the si-Parkin group (*p* < 0.01). The expression of USP30 (*p* < 0.01) protein was upregulated, while expression of optineurin (*p* < 0.05) and LAMP1 (*p* < 0.01) proteins were downregulated in the si-Parkin group (Figure 5B,C). No significant difference was observed for the ratio of LC3-II/I (*p* > 0.05; Figure 5F). These results suggested that knockdown of parkin resulted in a major loss of the protective effect of S3 on RGCs under excitotoxicity.

## 3. Discussion

These results demonstrated that S3 protected RGCs from NMDA-induced excitotoxicity and reinforced the parkin-mediated mitophagy pathway. S3 treatment improved mitochondrial health and promoted the Mfn2 ubiquitination and the OPA1 expression in RGCs. At the same time, the knockdown of parkin increased the USP30 expression and abolished the protective effect of S3 on RGCs under excitotoxicity.

Selective removal of damaged mitochondria via a process known as mitophagy is indispensable for mitochondrial quality control. Parkin, an E3 ubiquitin ligase, plays a pivotal role in mediating the mitophagy pathway [4,5,6]. Growing evidence has suggested that deubiquitinases may serve as a key negative regulator of parkin-mediated mitochondrial ubiquitination. In the current study, we provided in vitro evidence that, as a USP30 inhibitor, S3 promoted RGC survival against NMDA-induced excitotoxicity. The protein expression of parkin was upregulated in RGCs after S3 treatment under excitotoxicity, while knockdown of parkin diminished the protective effect of S3 on RGCs. The downstream pathway in parkin-mediated mitophagy was further investigated to elucidate the mechanism involved in the effect of S3 on RGC. When mitochondria became depolarized, parkin was activated and subsequently ubiquitinated outer mitochondrial membrane proteins. Then, autophagy receptors such as optineurin were recruited to the ubiquitinated mitochondria, followed by the formation of LC3-positive phagophores that degrade mitochondria via the lysosome [12]. Our data showed that the protein levels of LC3 and lysosome marker LAMP1 were upregulated following S3 treatment, while knockdown of parkin significantly decreased these proteins in RGCs with S3 treatment under excitotoxicity. All these data supported the assumption that S3 could protect RGCs through parkin-mediated mitophagy against excitotoxicity.

USP30 is an established mitochondrial outer membrane-localized deubiquitinase that negatively regulates parkin-dependent mitophagy through deubiquitylation of several outer mitochondrial membrane substrates, including Mfns [13]. Previously, Yue et al. reported that the inhibition of USP30 by S3 leads to an increase in non-degradative ubiquitination of Mfns, which promotes mitochondrial fusion in mouse embryonic fibroblasts [11]. Our data showed that the ubiquitination level of Mfn2 was increased after S3 treatment in RGCs under excitotoxicity, while there was no significant change in the protein level of Mfn2. Meanwhile, the increased colocalization of parkin with the mitochondria marker TOMM20 was observed in RGCs after S3 treatment. Interestingly, our data showed that knockdown of parkin significantly increased the protein level of USP30 in RGCs with S3 treatment. This supports the previous findings that parkin might directly degrade the intrinsic USP30, aside from the interplay between parkin and USP30 on mitochondrial membrane protein substrates [9]. These results indicated that the enhanced ubiquitination of Mfn2 induced by S3 treatment may facilitate the recruitment of parkin to damaged mitochondria in RGCs.

On the other hand, the fusion of mitochondria relies on the coordinated interaction of Mfns in the outer mitochondrial membrane and OPA1 located in the inner mitochondrial membrane. Fused mitochondria are thought to be more resistant to environmental stress [14]. Given the role of Mfn2 in mitochondrial fusion, one may assume that the increased ubiquitination of Mfn2 may enhance its activity and promote mitochondrial fusion in stressful RGCs. Our data, which indicate that mitochondria in RGC soma and axons become longer and highly interconnected, as well as the increased protein level of OPA1 following S3 administration, are suggestive of such an assumption. We previously reported that genetically upregulating OPA1 protects RGCs via enhancing mitochondrial fusion and parkin-mediated mitophagy [15]. In addition, OPA1 had been found to be a downstream mediator of the stress-protective activity of parkin [16]. Since a balance between mitochondrial dynamics and degradation is essential for mitochondria quality control within RGCs, further studies focusing on the interplay mechanism among Mfn2, OPA1 and parkin in RGCs under basal and stressful conditions are warranted.

In summary, our study identified a novel role for a naturally derived compound, S3, in protecting RGCs and improving mitochondrial quality control under excitotoxicity. The mechanism involved in the S3 effect on RGCs mainly occurred through enhancement of parkin-mediated mitophagy. The neuroprotective potential and clinic value of S3 in glaucoma and other neurodegenerative ocular diseases deserves further investigation.

## 4. Materials and Methods

### 4.1. RGCs Culture and Transfection

Primary RGCs were isolated and purified according to our modified protocol, as described in detail previously [17]. In brief, retinal tissues from three-day-old Sprague Dawley rats were cultured through a two-step panning procedure. RGCs were transfected with Lipofectamine RNAiMAX (Invitrogen, Carlsbad, CA, United States) for small interfering RNA (siRNA) experiments. The siRNA targeted against parkin was designed by GenePharma (Shanghai, China). The sequences designed were as follows: 5′-GGAACAACAGAGUAUCGUUTT-3′.

### 4.2. S3 Administration

S3 (provided by Prof. Liu Haiyang, Kunming Institute of Botany, Kunming, China) was dissolved in 3% DMSO, which was then used as the vehicle control. Three days after seeding, cultured RGCs were treated with different concentrations of S3 (1, 2, 3, 5 µM) or 3% DMSO for 24 h. Then, the RGCs were exposed to 100 μM NMDA (Sigma-Aldrich, St. Louis, MO, United States) for another 24 h in a 5% CO_2_ tissue culture incubator.

### 4.3. Measurement of Mitochondrial Membrane Potential, Cytotoxicity and Apoptosis

According to the protocol described in detail previously, the level of mitochondrial membrane potential in RGCs was measured using the JC-1 assay kit (Abcam, Cambridge, MA, United States). The level of cell damage was evaluated using both lactate dehydrogenase (LDH) assay kit (Beyotime Biotechnology, Shanghai, China) and Hoechst staining to count the percentage of apoptotic cells.

### 4.4. Immunoprecipitation

Immunoprecipitation was performed using Pierce Classic IP Kit (26146, Thermo Fisher Scientific, Waltham, USA) and under the manufacturer’s instructions. RGCs were lysed in IP Lysis (0.025 M Tris, 0.15 M NaCl, 0.001 M EDTA, 1% NP-40, 5% glycerol, pH 7.4) on the ice; then, the Mfn2 antibody was added, and the mixture was incubated overnight at 4 °C to form the immune complex. Protein A/G Plus Agarose was added, and the mixture was shaken gently at 4 ℃. Non-reducing Lane Marker Sample Buffer containing 0.02 M DTT was added to the mixture and then incubated at 100 °C for 5–10 min. The eluate was collected after centrifuging. After the samples cooled to room temperature, they were applied to the SDS-PAGE gel and transferred to PVDF membranes for Western blotting.

### 4.5. Immunofluorescence and Mitotracker Staining

After treatment, samples were fixed in 4% paraformaldehyde, washed with PBS 3 times and permeabilized with 0.3% Triton X-100 at room temperature for 20 min. Afterwards, samples were immersed in 5% BSA for 1 h and then incubated with primary antibodies such as LC3 (1:200; Abcam), LAMP1 (1:200; Abcam), anti-TOMM20 (1:1000, CST) and anti-parkin (1:200; Abcam) overnight at 4 °C. After several washes in PBS, Alexa Fluor 488-conjugated and 594-conjugated anti-rabbit IgG antibody (1:200; Life Technologies, Grand Island, NY, United States) was applied to capture primary antibodies at 25 ℃. Mitochondria were stained with 200 nM MitoTracker (Molecular Probes, Life Technologies) at 37 °C for 30 min according to the manufacturer’s instructions. Lastly, nuclei were dyed using Hoechst 33342 (1 μg/mL; Life Technologies) for 10 min. Photos were taken with a Leica SP8 confocal microscope.

### 4.6. Western Blot Analysis

RGCs were lysed using RIPA buffer (Beyotime Biotechnology, Shanghai, China) to obtain protein samples. Each sample (20 μg) was separated by SDS-PAGE gel and transferred to PVDF membranes. The membranes were incubated with several primary antibodies such as anti-parkin (1:200; Abcam), anti-optineurin (1:200; Abcam), anti-OPA1 (1:1000; Abcam), anti-LC3 (1:2000; Abcam), anti-LAMP1 (1:1000; Abcam), anti-USP30 (1:1000; Affinity), anti-MFN2 (1:1000, Abcam), anti-Ubiquitin Antibody (1:1000, LifeSensors) and anti-GAPDH (1:2000; Abcam) at 4 °C overnight and finally with peroxidase-conjugated anti-rabbit IgG (1:5000; Jackson). Images were developed with a chemiluminescence detection kit (SuperSignal™, Thermo Fisher) and analyzed by a 4000 MM PRO Kodak Image Station (Carestream, Rochester, NY, USA).

### 4.7. Statistical Analysis

Statistical analysis was processed by software Prism 8.0 (Graph Pad Software Inc., San Diego, CA, USA). All experiments were repeated at least 3 times. One-way analysis of variance and Tukey’s multiple comparisons test were performed to test the difference. Data are expressed as mean ± standard deviation. A *p*-value less than 0.05 was considered statistically significant.

## Figures and Tables

**Figure 1 molecules-27-04957-f001:**
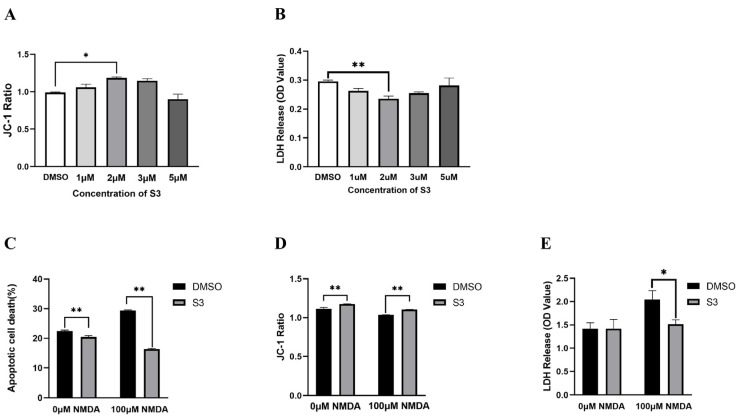
Effects of S3 on RGCs under basal and excitotoxic conditions. LDH and JC-1 assays showed that the 2 μM concentration of S3 is optimal for RGC viability (**A**,**B**). RGCs incubated with 2 μM S3 under excitotoxicity showed less apoptotic cell death (**C**), increased mitochondrial membrane potential (**D**) and less cytotoxic LDH release (**E**). *n* = 3, * *p* < 0.05, ** *p* < 0.01.

**Figure 2 molecules-27-04957-f002:**
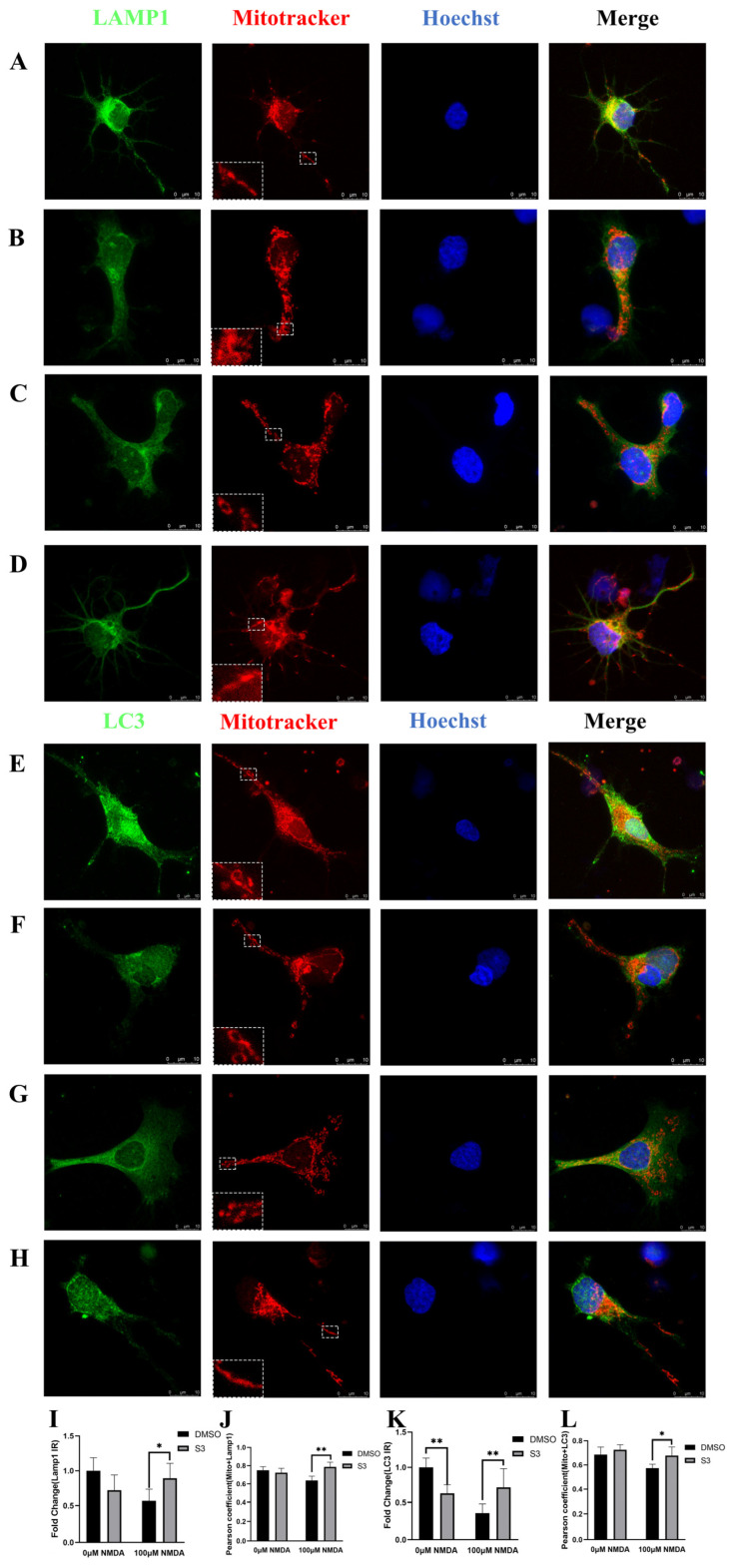
Immunofluorescence of RGCs co-stained with MitoTracker and LAMP1 or LC3. In vehicle-treated blank RGCs, a tubular mitochondrial network was distributed around the cell (**A**,**E**). In S3-treated blank RGCs, decreased immunoreactivity of LC3 was observed (**K**), while there was no significant change in LAMP1 expression (**I**) and mitochondria morphology (**B**,**F**). In vehicle-treated RGCs under excitotoxicity, small dot-like mitochondria were dispersed in the soma and axon (**C**,**G**). In S3-treated RGCs under excitotoxicity, elongated mitochondria with increased amounts were present around the nucleus and in the axon, and the immunoreactivity of LC3 and LAMP1 was upregulated under excitotoxicity (**D**,**H**). The levels of colocalization of LAMP1 and Mitotracker, LC3 and Mitotracker were increased in S3-treated RGCs, assessed by using the Pearson coefficient (Image J software) (**J**,**L**). *n* = 6, * *p* < 0.05, ** *p* < 0.01. Scale bar = 10 μm.

**Figure 3 molecules-27-04957-f003:**
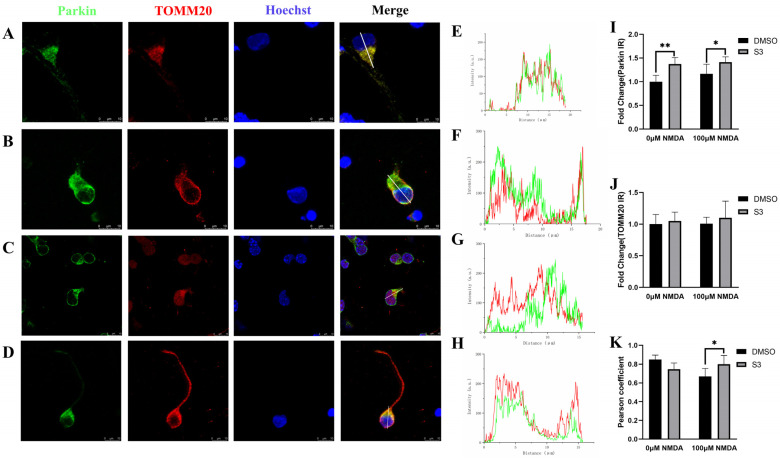
Immunofluorescence of RGCs co-stained with TOMM20 and parkin. The immunoreactivity of parkin was upregulated, while TOMM20 remained same under both S3-treated blank and excitotoxic conditions (**A**–**D**,**I**,**J**). The levels of colocalization of Parkin and TOMM20 were increased in S3-treated RGCs under excitotoxicity (**K**). Line graph densities represent the fluorescence intensity along the randomly selected section (white line). The consistent trend of the two curves indicated that parkin and TOMM20 tended to co-locate spatially under excitotoxicity (**E**–**H**). Line graph densities were quantified by Image J software. a.u., arbitrary units. *n* = 6, * *p* < 0.05, ** *p* < 0.01. Scale bar = 10 μm.

**Figure 4 molecules-27-04957-f004:**
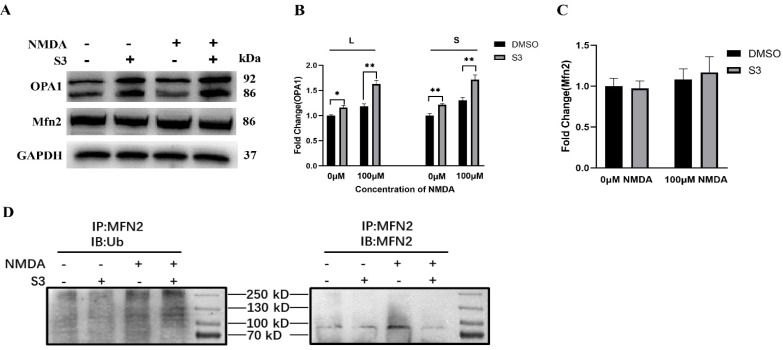
S3 upregulated the ubiquitination level of Mfn2 in RGCs under excitotoxicity. Compared with the vehicle-treated group, the expression of isoforms of L and S of OPA1 was increased in the S3-treated group (**A**,**B**). No change was observed in the protein level of Mfn2 (**A**,**C**). Lysates of RGCs were immunoprecipitated with an anti-Mfn2 antibody. The result showed that S3 treatment increased the ubiquitination level of Mfn2 in RGCs under excitotoxicity (**D**). *n* = 3, * *p* < 0.05, ** *p* < 0.01.

**Figure 5 molecules-27-04957-f005:**
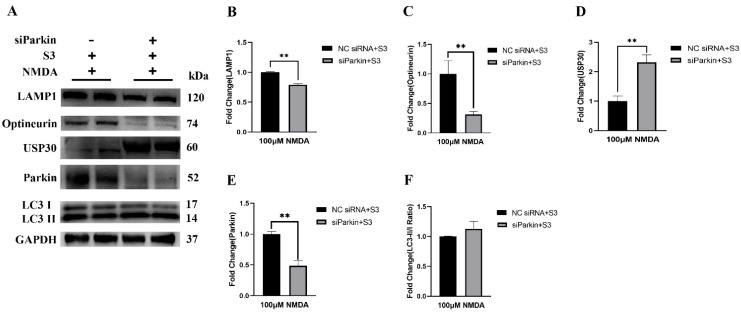
The effect of parkin knockdown on S3-treated RGCs under excitotoxicity. Compared with the control group, knockdown of parkin (**A**,**E**) increased the protein level of USP30 (**D**) and decreased the protein level of LAMP1 (**B**), optineurin (**C**). There was no change in the ratio of LC3-II/I (**F**). *n* = 3, * *p* < 0.05. ** *p* < 0.01.

## Data Availability

The data presented in this study are available on request from the corresponding author.

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
