# Peer review of "A Small Natural Molecule S3 Protects Retinal Ganglion Cells and Promotes Parkin-Mediated Mitophagy against Excitotoxicity"

_molecules, 2022, doi:10.3390/molecules27154957_

Round 1

Reviewer 1 Report

This study has interesting findings but requires adequate controls to be able to draw the conclusions that are presented in the manuscript. Lack of controls- particularly not using the 0uM NMDA condition for imaging experiments makes it a weak study design. It is difficult to assess the effects of S3 unless it can be shown that it’s effects are pertinent- meaning excitotoxicity in the cell culture experiments in your study perturbs the various measurements shown by imaging and western blots and that S3 can prevent them. 

Author Response

Changes Made in Response to the Reviewers´ Comments

(Re:  Molecules-1821966: A small natural molecule S3 protects retinal ganglion cells and promotes parkin-mediated mitophagy against excitotoxicity)

The authors would like to thank the reviewers for the constructive and helpful comments to improve the manuscript. We have responded to each of the suggestions raised by the reviewers and modified the manuscript where requested.

Reviewer 1

This study has interesting findings but requires adequate controls to be able to draw the conclusions that are presented in the manuscript. Lack of controls- particularly not using the 0uM NMDA condition for imaging experiments makes it a weak study design. It is difficult to assess the effects of S3 unless it can be shown that it’s effects are pertinent- meaning excitotoxicity in the cell culture experiments in your study perturbs the various measurements shown by imaging and western blots and that S3 can prevent them. 

Our response: Thanks for your valuable comment. We have added data using the 0uM NMDA condition for imaging experiments. Please see revised manuscript (Pg.3-5) and Fig. 2 & 3. In addition, the data on cell viability, mitochondria membrane potential and western blot using the 0uM NMDA condition were present on Fig. 1 and Fig.4.

Reviewer 2 Report

1.    The abstract looks more like a summary and lacks novelty. The authors should bridge the gap between the earlier published work and future directions. The abstract should briefly state the purpose of the research, the principal results, and major conclusions. An abstract is often presented separately from the article, so it must be able to stand alone.

2.    The authors are advised to provide a detailed account of small natural molecule S3.

3.    It is suggested to present the article’s structure at the end of the introduction. Comparing the present research results with similar studies done before is suggested.

4.     “Mammalian mitophagy could be mediated by an E3 ubiquitin ligase parkin that marks 31 defect mitochondria for degradation “. More evidence is required to support this statement.

5.     “It has been reported that S3 could block the deubiquitinases activity of USP30 39 and promote mitochondrial fusion”. More background studies should support this statement.

6.    The manuscript should be carefully revised, and the grammatical errors should be addressed.

7.    The authors are advised to improve the quality of the images if possible.

8.    The discussion section does not correctly address the important points and future perspectives. The author should add a few important points. A clear indication of how this work can benefit society should be provided.

Author Response

Changes Made in Response to the Reviewers´ Comments

(Re:  Molecules-1821966: A small natural molecule S3 protects retinal ganglion cells and promotes parkin-mediated mitophagy against excitotoxicity)

The authors would like to thank the reviewers for the constructive and helpful comments to improve the manuscript. We have responded to each of the suggestions raised by the reviewers and modified the manuscript where requested.

Reviewer 2

  1. The abstract looks more like a summary and lacks novelty. The authors should bridge the gap between the earlier published work and future directions. The abstract should briefly state the purpose of the research, the principal results, and major conclusions. An abstract is often presented separately from the article, so it must be able to stand alone.

Our response: Thanks for your valuable comment. The abstract has been revised according to your suggestion.

  1. The authors are advised to provide a detailed account of small natural molecule S3.
  2. It is suggested to present the article’s structure at the end of the introduction. Comparing the present research results with similar studies done before is suggested.
  3. “Mammalian mitophagy could be mediated by an E3 ubiquitin ligase parkin that marks 31 defect mitochondria for degradation “. More evidence is required to support this statement.
  4. “It has been reported that S3 could block the deubiquitinases activity of USP30 39 and promote mitochondrial fusion”. More background studies should support this statement.

Our response: Thanks! We have revised the introduction section according to your suggestion.

  1. The manuscript should be carefully revised, and the grammatical errors should be addressed.
  2. The authors are advised to improve the quality of the images if possible.

Our response: Thanks! We have improved the quality of the images and the grammatical errors in the manuscript.

  1. The discussion section does not correctly address the important points and future perspectives. The author should add a few important points. A clear indication of how this work can benefit society should be provided.

Our response: Thanks! The discussion section has been revised according to your suggestion.

Round 2

Reviewer 1 Report

In the updated manuscript, it was good to see sone of the suggestions taken into consideration. It would be an important advance to try and represent data not as just bar plots but to show the distribution going forward and is currently the most common way to represent data.

Reviewer 2 Report

Accept